# A potential tool for marine biogeography: eDNA-dominant fish species differ among coastal habitats and by season concordant with gear-based assessments

**Mark Y. Stoeckle**[1]*, **Jesse H. Ausubel**[1], **Greg Hinks**[2], **Stacy M. VanMorter**[2]

**1** Program for the Human Environment, The Rockefeller University, New York, New York, United States of America, **2** Bureau of Marine Fisheries, New Jersey Department of Environmental Protection, Port Republic, New Jersey, United States of America

\* mark.stoeckle@rockefeller.edu

**Data Availability Statement:** Analyzed data is included within the manuscript and in Supporting Information Files. In addition, Illumina FASTQ files

## Abstract

Effective ocean management asks for up-to-date knowledge of marine biogeography. Here we compare eDNA and gear-based assessments of marine fish populations using an approach that focuses on the commonest species. The protocol takes advantage of the "hollow curve" of species abundance distributions, with a minority of species comprising the great majority of individuals or biomass. We analyzed new and published teleost eDNA metabarcoding surveys from three neighboring northwest Atlantic coastal locations representing sandy, rocky, or estuary habitat. Fish eDNA followed a hollow curve species abundance distribution at each location—the 10 commonest taxa accounted for more than 90% of eDNA copies. Top ten taxa were designated eDNA-dominant species (eDDS) and categorized as habitat-associated (top 10 in one study) or as shared. eDDS by category were similarly abundant in concurrent bottom trawl and seine surveys. eDDS habitat category profiles correctly classified most (94%-100%) individual eDNA and capture measurements within surveys and recognized estuarine sites in other regional eDNA and seine studies. Using a category metric like that for habitats, eDDS demonstrated strong seasonal turnover concordant with trawl catch weights. eDNA seasonal profiles applied to historical trawl and seine records highlighted known long-term trends in mid-Atlantic fish populations. This study provides evidence that eDNA-abundant fish species differ among coastal habitats and by season consistent with gear-based assessments. Grouping abundant species by category facilitated comparisons among habitats and integration with established surveys. eDNA metabarcoding of dominant fish species potentially offers a useful tool for marine biogeography and ocean monitoring.

## Introduction

The growing anthropogenic footprint in the oceans potentially threatens marine life and human well-being. Impactful components include offshore facilities for wind power, oil and

underlying this manuscript are deposited in NCBI Bioprojects ID PRJNA638560, PRJNA793893, AND PRJNA864636, with links listed below. https://www.ncbi.nlm.nih.gov/bioproject/?term=PRJNA638560 https://www.ncbi.nlm.nih.gov/bioproject/?term=PRJNA793893 https://www.ncbi.nlm.nih.gov/bioproject/?term=PRJNA864636.

**Funding:** This work was supported by NOAA Grants NA20OAR01104XX (MYS,JHA), NA23OAR0110593 (MYS,JHA). https://www.noaa.gov/acquisition-grants The New Jersey Ocean Trawl Survey was supported by US Fish and Wildlife Service Wildlife and Sport Fish Restoration Program. https://www.fws.gov/program/sport-fish-restoration The Raritan Bay Seine Survey was supported by NOAA Interjurisdictional Fisheries Act of 1986 Grant NA20NMF4070069. https://www.noaa.gov/acquisition-grants.

**Competing interests:** The authors have declared that no competing interests exist.

gas extraction, aquaculture, sea-floor telecommunication equipment, commercial and recreational fishing and sea traffic, and run-off from coastal development [1, 2]. Harms may arise through direct removal of sea life, chemical, sound, and light pollution, and physical alterations in ocean temperature or circulation. Assessing and mitigating these threats asks for timely, locally relevant information on the abundance and distribution of marine life [3, 4]. Challenges include vastness of oceans in area and depth, uncertainty in classification and bounding of habitats, and expense of existing survey methods. Environmental DNA (eDNA) promises to boost ocean biomonitoring. As compared to gear-based technologies it is relatively inexpensive, harmless to the environment, and can be performed with modest equipment in diverse habitats [5, 6]. Limitations include inability to assess individuals (e.g., size, age, sex, health), uncertainty in relating eDNA abundance to organism abundance, and incomplete genetic reference libraries. eDNA can be analyzed by single-species assays or by metabarcoding, the latter targeting multiple species in a taxonomic group [7]. Northwest Atlantic coastal fish are good candidates for this multi-species approach, given mostly complete genetic libraries and primers with limited amplification bias among species. Marine eDNA investigations have often focused on presence/absence with an emphasis on detecting rarer species. Accumulating reports indicate fish species relative abundance is similar with eDNA methods including metabarcoding as compared to established census technologies [8–11]. Addition of copy number standards enables measuring eDNA absolute abundance, a further step towards censusing fish populations with eDNA data [12–15]. This progress raises the issue of how best to meld eDNA surveys with those obtained with other methods, one case of a general challenge for fisheries science [16, 17].

Over 70 years ago, Fisher *et al.* (1943) and Preston (1948) drew attention to species counts being highly skewed [18, 19]. For a given site and taxonomic group, a few species are abundant and the remainder are relatively rare, generating a "hollow curve" species abundance distribution (SAD). A hollow curve SAD appears universal among animals, plants, and microbes in terrestrial and marine environments [20–24]. This paper assesses marine fish SADs in coastal habitats using eDNA metabarcoding. We provide evidence that marine fish eDNA follows a hollow curve SAD consistent with that obtained by trawl and seine surveys and that eDNA-dominant species (eDDS) differ among habitats and by season. We show eDDS profiles can be applied to classify additional datasets including those obtained with established methods and look at potential utility of an eDDS-based approach in ocean monitoring and marine biogeography.

## Materials and methods

### Surveys, water sampling

Northwest Atlantic marine fish eDNA surveys analyzed herein are listed in Table 1 and locations shown in Fig 1. **New Jersey Bureau of Marine Fisheries Ocean Trawl Survey (NJOTS)**, in operation since 1988, conducts 30 or 39 trawls five times yearly from coastline out to 30 m isobath about 30 km offshore [25]. Between January 2019 and January 2020, 1 L water samples for eDNA analysis were collected at surface and depth prior to about one-fourth of trawls [10] (S4–S7 Tables). In addition, **NJOTS** catch records for January 1999 and 2009 were analyzed (S14 and S15 Tables). For **East River eDNA time series**, 1 L water samples were obtained weekly during March to June 2022 at shoreline of East River, a rocky tidal channel connecting Long Island Sound and New York Harbor [26] (S8 Table). **Raritan Bay Seine Survey (RBSS)** was carried out during April to November 2022 by New Jersey Bureau of Marine Fisheries (S9–S12 Tables). Seine sites were chosen by stratified random design along 36 km southern shoreline of Raritan Bay, the estuarine mouth of Raritan River (Fig 1). Seining was conducted

**Table 1. Marine fish surveys analyzed in this study.**

| Survey | Habitat type | Topography | Survey method | Reference |
|---|---|---|---|---|
| A. New Jersey Ocean Trawl Survey (NJOTS) | sandy | 4000 km$^2$ coastal ocean | eDNA, bottom trawl | [10, 15, 29] |
| B. East River eDNA time series | rocky | shoreline point | eDNA | this study |
| C. Raritan Bay Seine Survey (RBSS) | estuary | 36 km shoreline | eDNA, seine | this study |
| D. Barnegat Bay eDNA time series | estuary | shoreline point | eDNA | [27] |
| E. Connecticut Estuarine Seine Survey (CESS) | estuary | 150 km shoreline | seine | [28] |
| F. Long Island Sound Trawl Survey (LISTS) | sandy, muddy | 3000 km$^2$ ocean inlet | bottom trawl | [28] |

2 or 3 times monthly with an average of 6 hauls per seining day (total hauls, 118), and number of individuals per species were recorded. A single location was surveyed on a given day. 1 L water samples for eDNA analysis were collected on each seining day prior to deploying nets to minimize sediment collection (average samples per day, 3.3; total, 61). Due to absent or minimal catches in some individual hauls, catch and eDNA data were grouped by day and

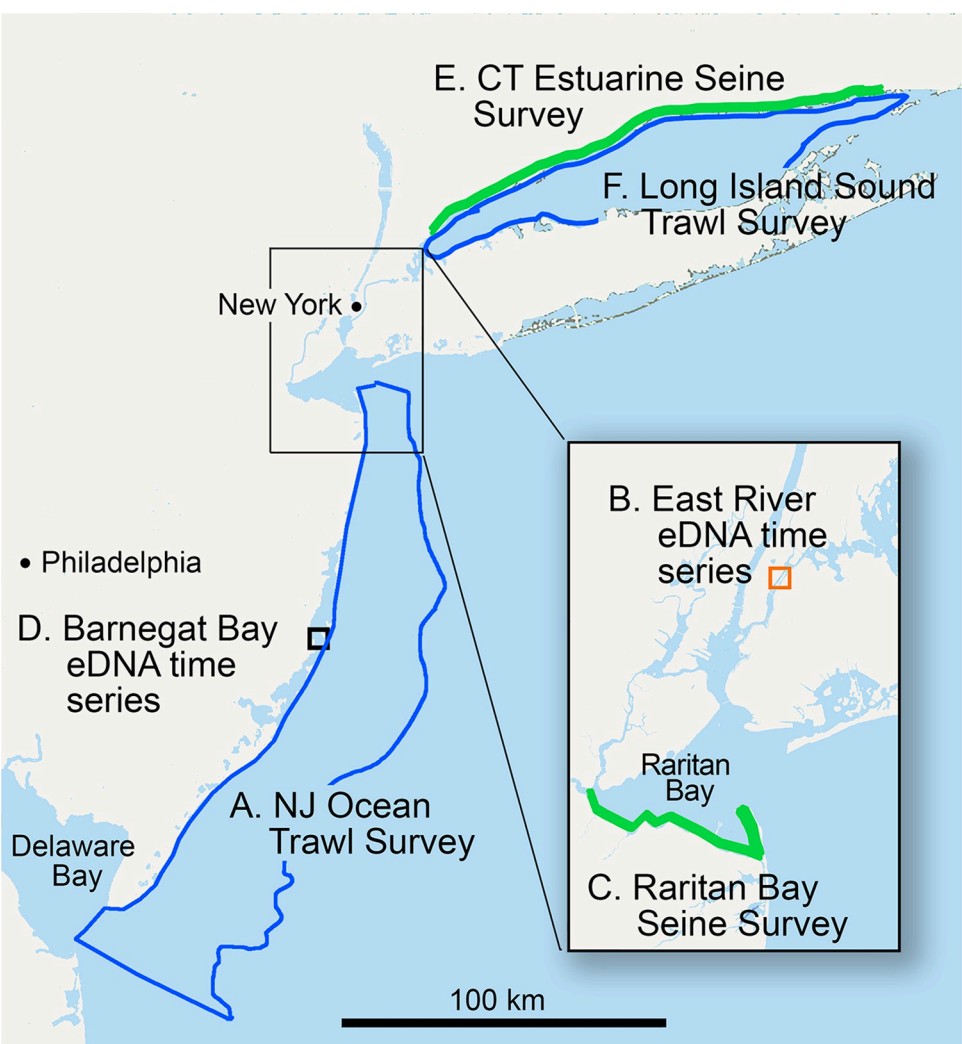

**Fig 1. Locations for surveys listed in Table 1.** Figure generated in Photoshop using USGS maps as template (https://apps.nationalmap.gov/viewer/).

normalized to values per haul and per volume filtered, respectively. For **Barnegat Bay eDNA time series**, 1 L water samples were collected at a single shoreline site approximately twice monthly from April 2017 to March 2019 (S13 Table) [27]. **Connecticut Estuarine Seine Survey (CESS)** and **Long Island Sound Trawl Survey (LISTS)** are conducted by Connecticut Department of Energy and Environmental Protection (DEEP) [28]. Survey protocols and catch data were downloaded from DEEP public website (https://portal.ct.gov/deep/fishing/fisheries-management/long-island-sound-trawl-survey). **CESS** monitors eight locations along about 150 km of northern shore of Long Island Sound once yearly in September. For this investigation, we compiled yearly total individuals per species for period 1988–2022 (S16 Table). **LISTS** conducts approximately 40 trawls per month during spring (April, May, June) and fall (September, October) at randomly selected sites in the Sound. For this study, we compiled yearly catch weights per species for period 1992–2022 (S17 Table). No animals were housed or experimented upon as part of this study. No endangered or protected species were collected during water sampling. NJOTS trawls that captured protected species were covered by NOAA incidental take permit number NER2018-14763.

## Water filtration, DNA procedures, bioinformatics

For all eDNA surveys, water samples were stored at 4° C and filtered within 24 h, or frozen at -20° C for up to 2 months until processing. Filtration was performed using wall suction and a glass manifold holding a 4.5 cm, 0.45 μM pore size nitrocellulose filter. If incomplete at 2 h, filtration was stopped and processed volume recorded. Filter clogging was limited to Raritan Bay Seine Survey samples, which yielded an average filtered volume of 560 mL (range, 250 mL—950 mL) (S10 Table). Filters were folded and stored in 15 mL tubes at -20° C. Negative filtration controls consisted of 1L samples of laboratory tap water processed using same equipment and methods as for field samples. DNA was extracted with Qiagen PowerSoil Pro Kit, eluted in 100 μL Buffer C6, and DNA yields checked with Qbit. Samples were amplified with TaKaRa High Yield PCR EcoDry™ Premix in 8-well strips, 5 μL DNA sample or reagent-grade water, and 200 nM Illumina-tailed Riaz primers complementary to mitochondrial-encoded 12S ribosomal gene (12S) [30] in 25 μL total volume (additional details S1 Table). Where noted, to calculate eDNA copies, amplifications were spiked with an ostrich 12S DNA amplicon standard which contains Riaz 12S primer binding sites identical to those in teleost fish [15, 29]. 5 μL of each reaction mix were run on a 2.5% agarose TBE gel with SyberSafe dye to evaluate amplification, and the remainder diluted 1:20 in Buffer EB (Qiagen). 5 μL of diluted product was indexed using Nextera XT Kit with Cytiva PuReTaq Ready-To-Go PCR beads (S1 Table). Following PCR, 5 μL reaction mix was run on gel with SyberSafe dye to assess indexing. Remainder of reaction mix was pooled with other libraries and cleaned with AMpure beads at 1:1. Library pools were adjusted to 10 ng/μL and sent to GENEWIZ for sequencing on Illumina MiSeq 150 bp x 2. Dereplicated FASTQ files were loaded into a DADA2 bioinformatic pipeline [29]. As previously described, to reduce assignment errors due to tag jumping, DADA2 output tables were filtered by excluding ASV detections comprising less than 0.1% of the total reads for that ASV among all libraries in the run. Laboratory tap water eDNA and reagent-grade water libraries were negative for fish ASVs after filtering. ASVs were identified by matching to an internal library (S2 Table, S1 File) and by manually searching GenBank; 100% matches were retained for further evaluation. eDNA copies were calculated using internal ostrich standard (29). For Raritan Bay samples, eDNA copies were adjusted to account for filtered volume. ASVs matching cartilaginous and freshwater species were excluded. Some trawl and seine catch records were lumped to match eDNA identifications, as targeted 12S amplicon does not distinguish some regional fish species (S3 Table). Scientific names are given in Supporting Tables.

## Species abundance distributions

The NJOTS, East River, and RBBS surveys described above were considered to represent sandy, rocky, and estuary coastal habitat, respectively (Table 1A–1C; Fig 1). We identified the ten most abundant taxa in terms of eDNA copies per liter in each study as a whole. These were designated eDNA-dominant species (eDDS) and categorized as habitat-associated (top ten one survey only) or as shared. The resulting sets of species defined "eDDS habitat profiles". The first rationale for abstracting the data in this way was that the top ten species accounted for the great majority of eDNA and presumably dominated ecosystem processes. The second was that top ten species tended to be similar within and differ among studies. The overarching goal was to analyze large-scale differences among habitats. eDDS habitat profiles based on the three surveys were applied to individual eDNA observations, concurrent trawl and seine records, and historical records from other locations. To reiterate, eDDS were defined by aggregate absolute abundance in a survey as a whole and then applied to evaluate proportional abundance in individual samples and other field studies. We analyzed proportional abundances as earlier work showed large differences among individual samples and trawls in absolute eDNA copies/reads and catch weights, respectively (e.g., S2 Fig) [10]. Absolute values for eDNA and trawl and seine assessments are included in Supporting Tables. We set two provisional criteria to assess differences within and among studies. First, dominance, i.e., did same-habitat-associated plus shared eDDS comprise greater than 80% of fish eDNA/catch? Second, classification, namely, was proportion of same-habitat-associated greater than that of other-habitat-associated fish eDNA/catch? In addition to comparisons by habitat, methodology described above generated seasonal eDDS profiles based on winter and summer eDNA SADs in NJOTS. These seasonal profiles were applied to individual water samples, to concurrent trawl data, and to historical trawl records from NJOTS and LISTS. Statistical tests were done in GraphPad Prism 10.2.2. Leave-one-out meta-analysis was performed by omitting one sample or seine day at a time from each habitat or seasonal eDNA dataset then re-calculating the eDDS profile based on truncated set.

## Results

### eDNA species abundance distributions in coastal habitats

Bony fish eDNA SADs in the three regional surveys followed a hollow curve distribution, with a small number of abundant fish and a larger array of uncommon taxa (Fig 2) (S5, S8 and S12 Tables). The top ten taxa, designated eDNA-dominant species (eDDS) (see Materials and methods), comprised a minority (16%-25%) of fish species but nearly all (92%-97%) bony fish eDNA. eDDS were categorized as habitat-associated (top ten in one survey) or as shared among surveys. Most individual water samples and seine days met dominance and classification criteria described in Materials and methods (Fig 3). As described above, for NJOTS, each water sample corresponds to a single trawl location within 4,000 km$^2$ survey area and for RBBS, each seine day represents a single location along 36 km shoreline. Same-habitat-associated plus shared eDDS comprised greater than 80% eDNA in nearly all (48/49; 98%) individual water samples/seine days, and same habitat-associated eDDS were more abundant than other habitat-associated taxa in nearly all individual water samples/seine days (48/49; 98%) (probability that classifications did not differ among locations, p < 0.0001, Fisher's exact test; probability that a given sample/seine day was misclassified, 2%, 95% confidence interval 0%-12%, Wald's method). Similar results were obtained when Raritan Bay water samples were analyzed individually rather than by seine day: 85% met dominance criterion and 90% were correctly classified (S1 Fig). As described in Materials and methods, leave-out-one meta-analysis

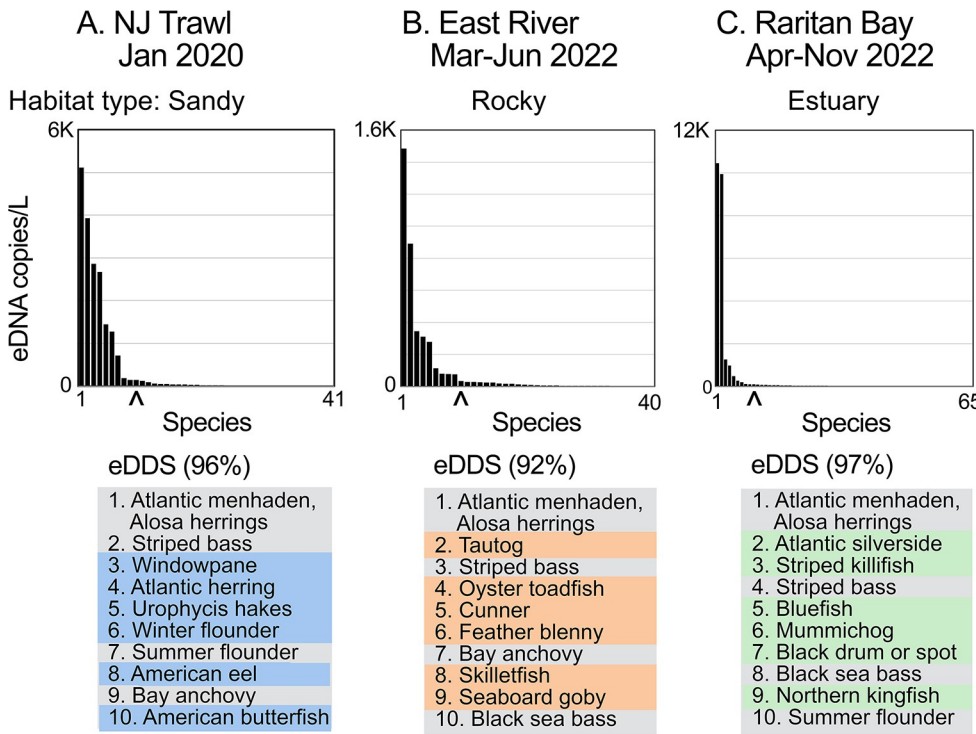

**Fig 2. Fish eDNA habitat SADs.** Commonest fish species in mid-Atlantic surveys A, B, and C representing sandy, rocky, and estuary habitat, respectively (Table 1 and Fig 1). Each column corresponds to one species and species are ranked by decreasing aggregate abundance measured as eDNA copies/L. Scale differs between surveys. Caret (^) marks 10th most common taxon. At bottom, proportion total bony fish eDNA occupied by top ten is shown and top ten fish for each survey are listed by decreasing eDNA abundance. Habitat-associated eDDS as defined in Materials and methods are highlighted in color (blue, orange, or green) and shared eDDS are highlighted gray.

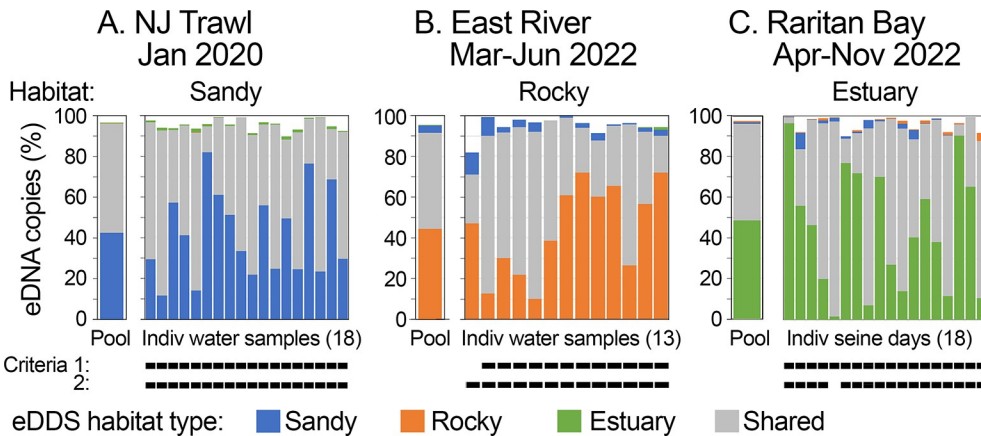

**Fig 3. eDDS habitat profiles applied to pooled and individual water samples/seine days.** eDDS categories and colors are as shown in Fig 2. Pool proportions for each survey are calculated from aggregate eDNA copies/L. A gap at top of a column represents non-eDDS taxa. Criteria 1 and 2 represent dominance and classification metrics, respectively, described in Materials and methods. A dark bar indicates criterion was met.

assessed contribution of individual samples/seine days to eDDS habitat profiles. The top ten taxa were identical to those in the full dataset in most cases [16/18 (89%), 13/13 (100%), and 11/18 (61%), in sandy, rocky, and estuary datasets, respectively]; exceptions differed at 10th most common taxa.

## eDNA-dominant species in concurrent trawl and seine surveys

eDDS habitat profiles defined above were applied to accompanying trawl and seine surveys (Fig 4) (S4 and S11 Tables). Catch proportions in pooled data and most individual trawls/seine days met dominance and classification criteria described above. In particular, same-habitat-associated plus shared eDDS comprised greater than 80% catch in pooled data and in most individual trawls (26/30; 87%) and seine days (16/18; 89%). Pooled data and all individual trawls (30/30, 100%) and most (17/18; 94%) seine days were correctly classified, namely, proportion same-habitat-associated was greater than that of other-habitat-associated eDDS. Comparable agreement between eDNA and catch records was observed for individual species (Fig 5) (S6 Table).

## eDDS habitat profiles applied to additional locations

eDNA habitat profiles defined above recognized estuaries at other locales (Fig 6). In 2-year Barnegat Bay eDNA study, estuary-specific plus shared taxa accounted for more than 80% of bony fish eDNA in most (45/59, 75%) water samples, and nearly all (58/59, 98%) samples showed predominance of estuary-associated taxa (S13 Table) [27]. In 35-year CESS, estuary-specific plus shared taxa accounted for more than 80% of bony fish individuals in most years (30/35; 86%), and profiles correctly classified all years as estuarine (S16 Table). eDDS profiles highlighted the known long-term decline in winter flounder (*Pseudopleuronectes americanus*) in Long Island Sound, evident in Fig 6 as decreasing proportion of sandy habitat taxa in CESS [31, 32].

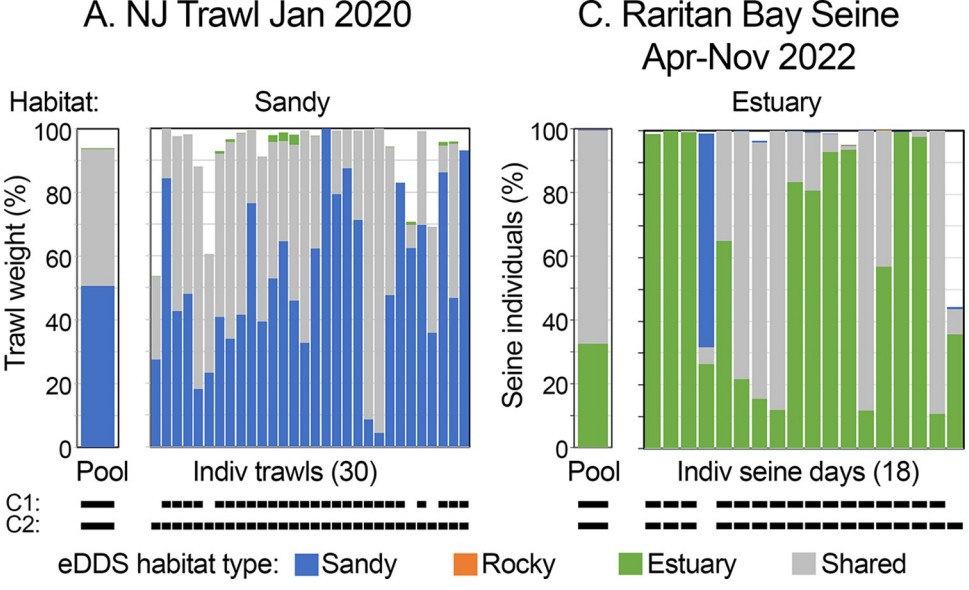

**Fig 4. eDDS habitat profiles applied to concurrent gear-based surveys.** A gap at top of a column represents non-eDDS taxa. As with eDNA data, pool proportions reflect aggregate weight or individuals for a survey (S4 and S11 Tables). C1 (criteria 1) and C2 (criteria 2) represent dominance and classification metrics described in Materials and methods.

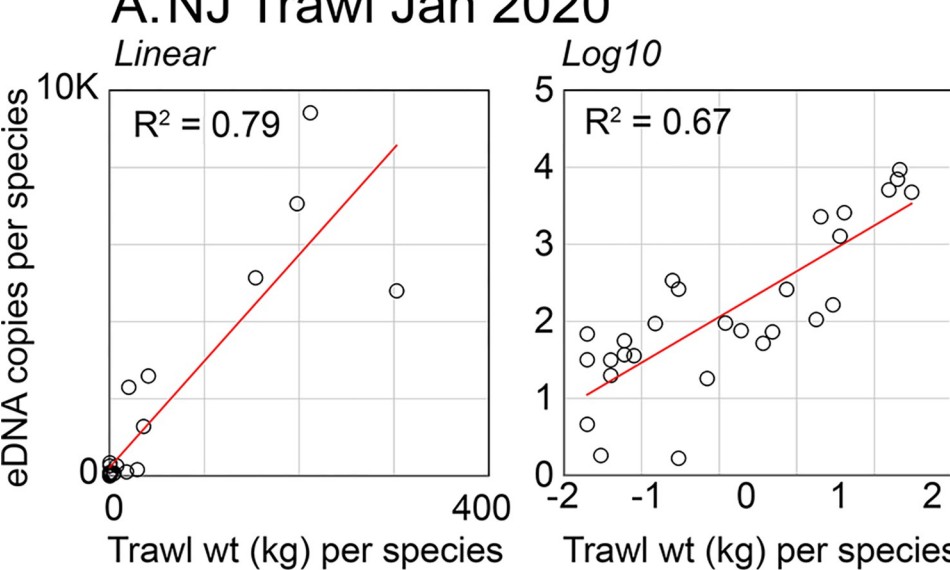

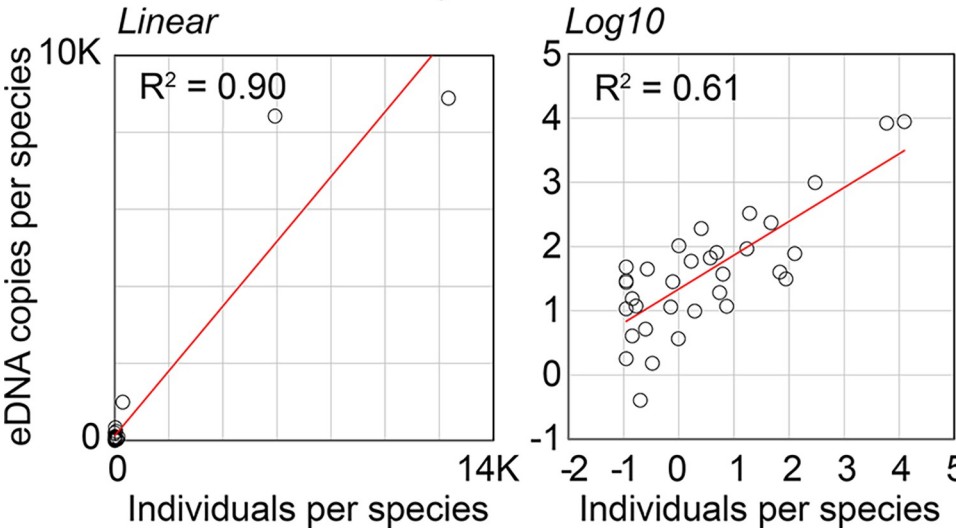

**Fig 5. eDNA copies vs trawl and seine catches.** Each circle represents 1 bony fish species. For NJOTS, values represent total catch weight or total eDNA copies. For RBSS, values represent total individuals per day (normalized to #seines/day) and total eDNA copies per day (normalized to #water samples/day).

### eDDS seasonal profiles

An eDDS protocol was applied to New Jersey Ocean Trawl Survey winter-summer phenology. This evaluation utilized eDNA reads due to absent information on copies for August 2019 water samples. In support of this substitution, copies/species were closely proportional to reads/species in NJOTS January 2020 data (S3 Fig). Seasonally-dominant species by eDNA demonstrated nearly complete turnover between winter and summer, and proportions were concordant with seasonal trawl weights (Fig 7) (S4, S5 and S7 Tables). Regarding dominance and classification criteria, same-season-specific plus shared eDDS comprised greater than 80%

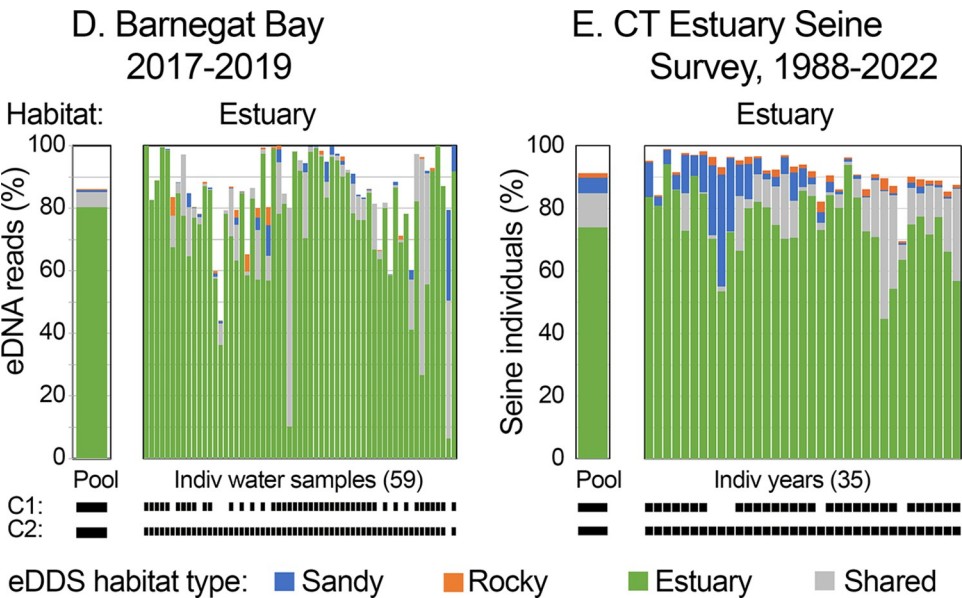

**Fig 6. eDDS habitat profiles applied to other locations ([Table 1](), [Fig 1]()).** At left, 2-year eDNA time series from Barnegat Bay, an estuary 80 km south of Raritan Bay. At right, 35 years of catch records from Connecticut Estuary Seine Survey conducted along northern shore of Long Island Sound, about 100 km northwest of Raritan Bay.

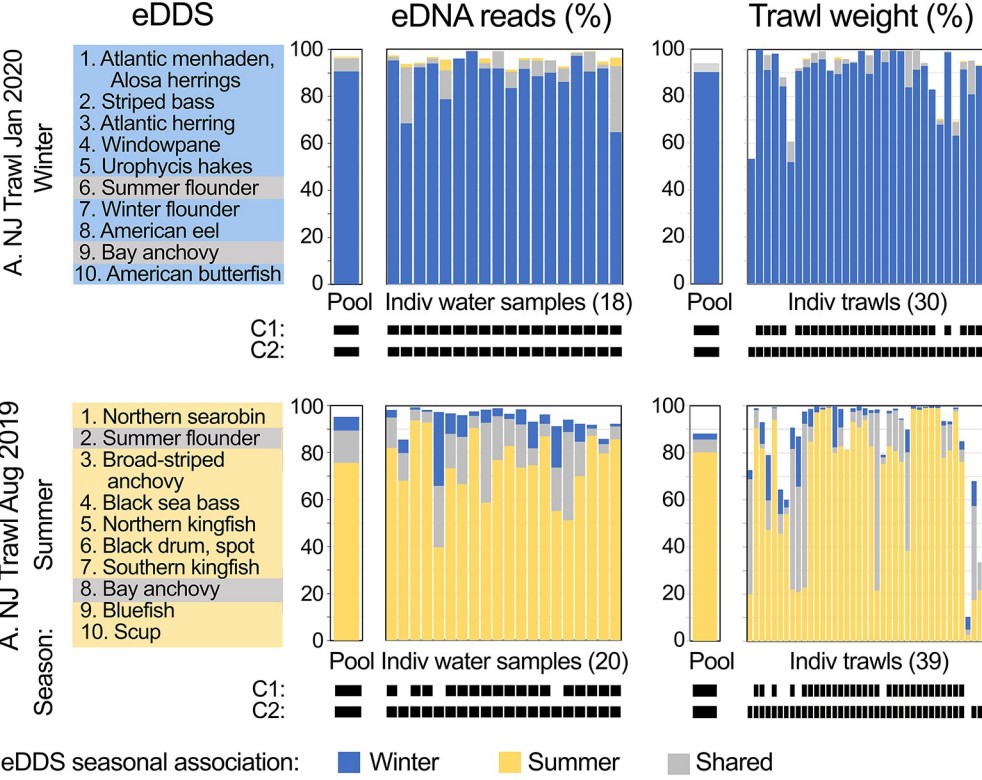

**Fig 7. eDDS protocol applied to NJOTS phenology.** At left, seasonal eDDS based on eDNA reads for January 2020 and August 2019. Colors highlight season-associated and shared taxa as indicated at bottom. In middle and right, eDDS seasonal categories applied to pooled and individual water samples and trawls. As with habitat comparisons, pool proportions represent aggregate reads or catch weight for a survey.

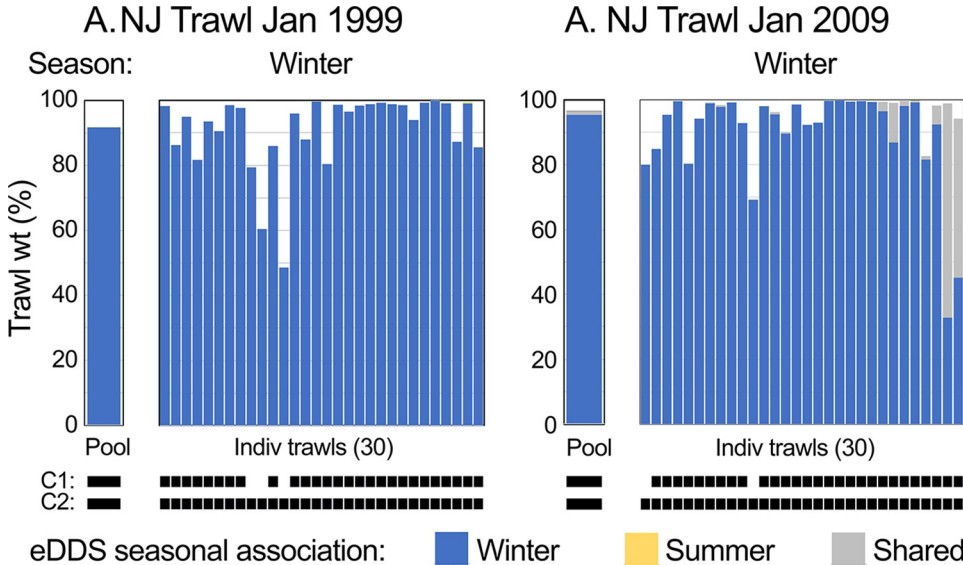

**Fig 8. NJOTS eDDS seasonal profiles applied to historical NJOTS records.** Compare to NJOTS January 2020 proportions in Fig 7.

eDNA reads/trawl weight in most individual water samples (35/38, 92%) and trawls (56/69, 81%). All water samples (38/38, 100%) and nearly all trawls (68/69, 99%) were correctly classified by season-specific eDDS. Leave-out-one meta-analysis assessed contribution of individual samples to eDDS seasonal profiles based on reads. The top ten taxa in truncated datasets were identical to those in the full set in most cases [13/20 (65%) and 14/18 (78%) in summer and winter datasets, respectively]; exceptions differed at 10th most common taxa.

## eDDS seasonal profiles applied to historical records

To further explore potential utility of eDDS metrics for evaluating long-term trends, eDDS seasonal profiles described above were applied to historical NJOTS and LISTS records. In NJOTS, winter-associated eDDS accounted for great majority of pooled catches and individual trawls in winter across two decades (92%, 95%, and 90%, in 1999, 2009, and 2020, respectively) (Figs 7 and 8) (S4, S14 and S15 Tables). There was a modest increase in shared taxa, evident in pooled data and individual trawls. In all instances this represented summer flounder (*Paralichthys dentatus*). Summer flounder were captured in 0%, 22% and 73% of trawls in January 1999, 2009, and 2020, respectively (probability detections unchanged from 1999 to 2009, p <0.0001, and from 2009 to 2020, p = 0.0182, Fisher's exact test). These findings were consistent with successful re-building of *P. dentatus* mid-Atlantic stocks, northward shift of *P. dentatus* populations beginning in 1960s, and long-term changes in fish distributions in Northeast U.S. shelf ecosystem [33–35]. Seasonal eDDS profiles based on NJOTS highlighted an increasing proportion of summer-associated species over past 30 years in Long Island Sound, well-documented in published reports (Fig 9) (S17 Table) [31, 32].

## Discussion

Here we analyzed marine bony fish eDNA species abundance distributions in northwest Atlantic coastal surveys. In each case, teleost eDNA followed a hollow curve SAD, "one of ecology's oldest and most universal laws" [36]. We explored differences in species composition among locations and by season. Comparisons focused on the ten commonest taxa by eDNA,

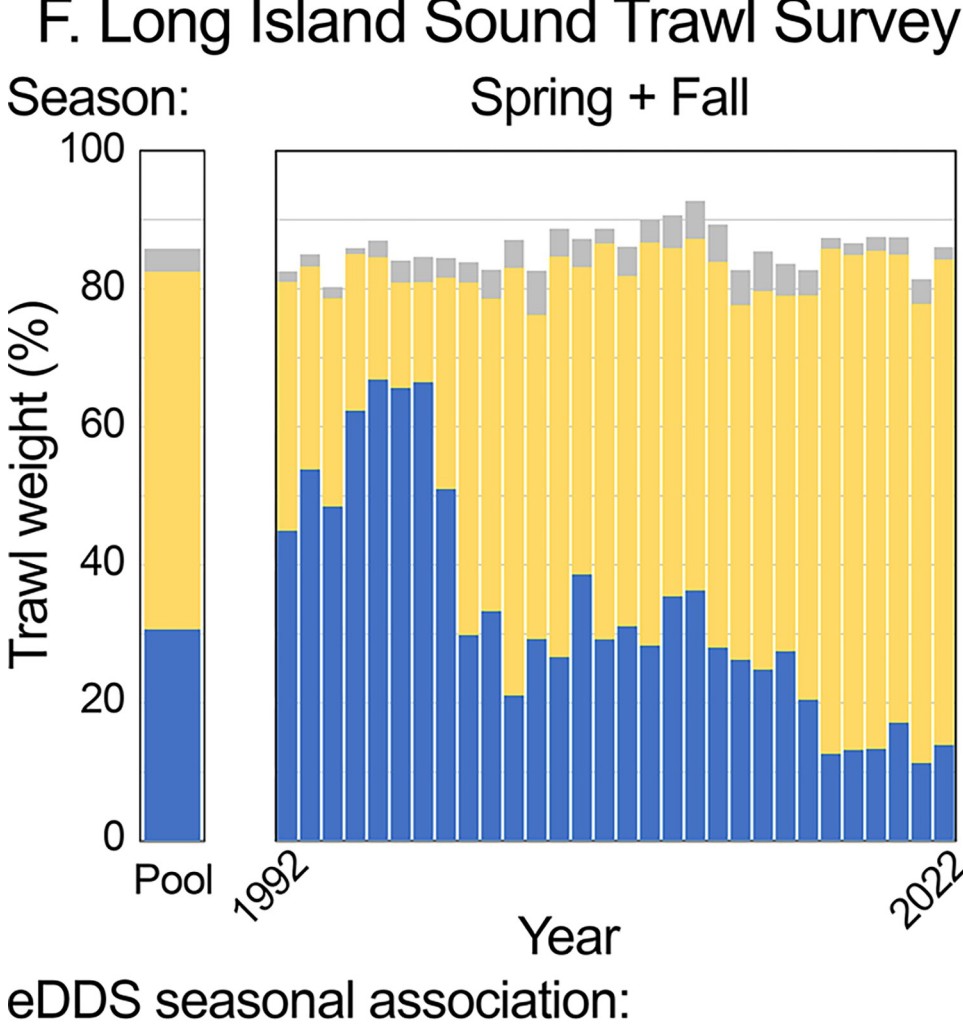

**Fig 9. NJOTS eDDS seasonal profiles applied to Long Island Sound Trawl Survey.**

categorizing these as habitat- or season-associated or shared. eDNA-dominant species differed among neighboring coastal locales and by season and were similarly manifest by eDNA and established capture methods. eDDS profiles recognized estuaries in eDNA and seine surveys conducted at other regional locations. Seasonal eDDS profiles applied to more than 30 years of trawl and seine records highlighted both stable and changing aspects of coastal fish populations. To recap, the commonest fish species differed among marine habitats and by season and were similarly evident by eDNA metabarcoding and gear-based technologies. Our findings support incorporating eDDS assessments in marine survey programs, with potential benefits to ocean biomonitoring and biogeography [37–39].

## Limitations

Limitations to dominant species methodology described here include that categories group species, which may obscure ecologically important differences between similar locations or

over time [40]. eDNA is subject to dispersal by settling, diffusion, and currents and by decay, processes that may introduce local variability [41, 42]. Nonetheless we found categorical SADs relatively uniform across large areas (NJOTS and RBBS cover about 4,000 km$^2$ coastal ocean and 36 km shoreline, respectively). Metabarcoding assays are subject to PCR bias, resulting in some taxa amplifying more efficiently than others, distorting apparent eDNA relative abundance [43]. Species-specific adjustments in PCR data may yield greater precision, particularly important for stock assessment of commercial taxa [14, 43]. Designating "top ten" as eDDS is an arbitrary cut-off that incorporates rare taxa in some datasets and not others. Potential alternative heuristics defining common species include the rank-ordered set comprising greater than 90% of total eDNA copies, or all individual species comprising greater than 1% eDNA copies. The extent to which eDNA distribution and fish distribution conform to physical habitat boundaries likely differs by species and life stage. For instance, black sea bass (*Centropristis striata*) is a structure-associated species [44]. However, adults were relatively abundant in summer NJOTS by eDNA and capture, even though trawls avoid structured areas, and juveniles are seasonally present in mid-Atlantic estuaries [45]. By design, a dominant species approach ignores rare species, a major focus of biodiversity study, including that directed at threatened or invasive organisms. This paper examined eDDS in surveys conducted in a priori defined habitats. Criteria that flag samples as likely representing new environments would aid ocean exploration and biogeography. The intensity of sampling needed to characterize eDNA-dominant species is unknown and may differ among locations. Other limitations arise from eDNA methods. Riaz 12S gene metabarcoding primers yield incomplete species resolution and work poorly to recover chondrichthyan eDNA; these problems could potentially be circumvented with alternate primers [27, 46]. More generally, all fish metabarcoding primers have some degree of amplification bias; analyzing samples with multiple primer sets is often advised to add confidence in species detection and abundance assessments [47]. Guri and colleagues recently demonstrated that species-specific adjustments that take into account PCR bias and fish catchability enable close prediction of absolute trawl catches from eDNA metabarcoding data [14]. At present, wide usage of this approach may be limited by need to determine PCR and catchability adjustment parameters for each species of interest in a particular environment. We suggest that the eDDS protocol described here, which takes advantage of the lopsided nature of SADs, is suitably accurate in present form for assessing dominant species. The protocol potentially offers a way to screen eDNA and traditional datasets for large-scale differences in common species by geography, season, or over time. Looking ahead, biorepositories of archived eDNA samples, advances in DNA methods, and more complex modelling than outlined here may yield greater utility in ocean biomonitoring. It may be desirable to utilize eDDS metabarcoding for taxa other than ichthyofauna. To our knowledge, to date fish are the only large marine metazoan group with limited-bias metabarcoding primers and substantiation that eDNA is generally proportional to biomass (see references in Introduction). Nonetheless, one might map marine biogeography by adopting a standardized metabarcoding assay that targets invertebrate taxa, for example [48, 49].

This investigation examined a small number of studies conducted in a single coastal region. Of particular interest for further work are marine locations that differ greatly in fish abundance, diversity and physical oceanography, such as pelagic and deep ocean and tropical reefs. Two lines of evidence suggest generalizability. First, extensive ecology literature beginning in the 1940s demonstrates hollow curve SADs are fundamental, i.e., "every community shows a hollow curve or hyperbolic shape on a histogram with many rare species and just a few common species" [36]. Mechanisms that underlie the ubiquity of skewed SADs are unknown. Mathematical modelling may provide insight [50, 51]. Log-series and log-normal distributions, first proposed by Fisher *et al.* (1943) and Preston (1948), respectively, remain primary

candidates [18, 19]. Connolly and colleagues analyzed over 1,000 SADs from "14 marine ecosystems ranging from intertidal habitats to abyssal depths, and from the tropics to polar regions" [24]. They found SADs were more consistent with log-normal distribution rather than with distributions predicted by neutral models [52]. A second line supporting generalizability is that, where studied, SADs are stable over time and uniform in space. For instance, copepods in the North Pacific Central Gyre were without significant changes in species rank order or percent abundance over 16 years and across distances up to 800 km [22, 53]. In the same open ocean region, compositions of larval and adult fish showed nearly constant species proportions over 5 years [20, 21].

The eDDS approach asks "what are the most abundant fish in this area?" and uses that information to characterize and compare survey locations. The output is an ecosystem snapshot depicting the common species that make up the great majority of biomass. eDDS proportions by category in individual observations were most often roughly consistent with pooled proportions, and dominant species profiles were mostly unchanged in leave-one-out meta-analysis, suggesting that modest collection effort may suffice to characterize a habitat at this level of resolution. Although beyond the scope of this investigation, sampling requirements could be further explored by separating observations into training and test sets. Established β-diversity metrics generate a numerical index of differences in species composition between observations (e.g., Bray-Curtis, Sorenson, Canberra) [54–58]. To evaluate variation among sites, NMDS and related methods represent β-diversity values in multi-dimensional virtual space with arbitrary scaling [e.g., 40]. In contrast, eDDS metrics compare arithmetic relative abundances, and eDDS profiles classify novel datasets without re-analyzing original data.

If supported by further investigation, eDDS profiles may offer a biological method to classify and bound ocean water masses. A vexing problem in ocean exploration is whether the biology of a region is unprecedented or familiar. An eDDS approach might quickly give a tentative answer to this question with comparisons to observations in the Ocean Biodiversity Information System (https://obis.org). With additional development, eDDS methodology might also offer ways to monitor climate change, impacts of wind farms and other structures, and consequences of creation of marine protected areas. A potentially productive line of enquiry is comparing fish eDDS-defined habitats to existing marine biogeographic demarcations, which are variously based on political boundaries, physical ocean parameters, and organism ranges particularly those of arthropods, molluscs, and other relatively sessile organisms [59–63]. Dominant species profiles shine a light on highly abundant species, which dominate ecosystem processes; further investigation may give insight into an ecological community as a whole: "The key to understanding the distribution of abundances in communities. . .may lie as much in understanding what characteristics of common species allow them to remain so abundant. . .as in understanding the dynamics and persistence of rare species" [24].

## Conclusion

The eDDS approach described here holds promise for aiding ongoing efforts characterizing marine ecosystems and aligns with recent U.S. national policy initiatives [64, 65]. Aquatic eDNA performs well in multiple settings that would otherwise require specialized equipment. Additional values to an eDDS protocol include relatively low cost of sample collection and analysis, absent environmental harm, and ease of integration with established technologies. Surveys focused on the most plentiful taxa require less effort than initiatives to census all resident fauna. Incorporating an eDDS-type approach in marine surveys may help classify ocean habitats at a range of scales and boost ongoing biomonitoring efforts for ecosystem-based management [66, 67].

## Supporting information

**S1 Table. PCR primers, parameters.**
(XLSX)

**S2 Table. Riaz 12S amplicon reference sequences.**
(XLSX)

**S3 Table. Riaz 12S amplicons shared among species.**
(XLSX)

**S4 Table. New Jersey Ocean Trawl Survey (NJOTS) Aug 2019, Jan 2020 fish catch weights.**
(XLSX)

**S5 Table. NJOTS Jan 2020 eDNA copies, reads.**
(XLSX)

**S6 Table. NJOTS Jan 2020 eDNA copies vs catch weight individual species.**
(XLSX)

**S7 Table. NJOTS Aug 2019 eDNA reads.**
(XLSX)

**S8 Table. East River time series 2022 eDNA copies.**
(XLSX)

**S9 Table. Raritan Bay Seine Survey (RBSS) 2022 physical data.**
(XLSX)

**S10 Table. RBSS 2022 water samples.**
(XLSX)

**S11 Table. RBSS 2022 seine catch individuals.**
(XLSX)

**S12 Table. RBSS 2022 eDNA copies.**
(XLSX)

**S13 Table. Barnegat Bay time series 2017–2019 eDNA reads.**
(XLSX)

**S14 Table. NJOTS Jan 1999 trawl weights.**
(XLSX)

**S15 Table. NJOTS Jan 2099 trawl weights.**
(XLSX)

**S16 Table. 1988–2022 CT Estuarine Seine Survey (CESS) seine catch data.**
(XLSX)

**S17 Table. 1992–2022 Long Island Sound Trawl Survey (LISTS) trawl catch data.**
(XLSX)

**S1 Fig. eDNA copies/L for Raritan Bay Seine Survey by individual water sample.**
(DOCX)

**S2 Fig. Absolute values for eDNA and capture data in January 2020 NJOTS and Raritan Bay surveys.**
(DOCX)

**S3 Fig. eDNA copies vs reads for NJOTS January 2020.**
(DOCX)

**S1 File. Riaz 12S amplicon reference sequences.**
(TXT)

## Acknowledgments

At The Rockefeller University we thank Jeanne Garbarino and Jen Bohn for laboratory resources and assistance. We thank Jason Adolf, Anne Bucklin, and Sam Chin for helpful discussions.

## Author Contributions

**Conceptualization:** Mark Y. Stoeckle, Jesse H. Ausubel, Greg Hinks.

**Data curation:** Mark Y. Stoeckle, Stacy M. VanMorter.

**Formal analysis:** Mark Y. Stoeckle, Jesse H. Ausubel, Stacy M. VanMorter.

**Funding acquisition:** Mark Y. Stoeckle, Jesse H. Ausubel, Greg Hinks.

**Investigation:** Mark Y. Stoeckle, Jesse H. Ausubel, Greg Hinks, Stacy M. VanMorter.

**Methodology:** Mark Y. Stoeckle, Jesse H. Ausubel, Greg Hinks, Stacy M. VanMorter.

**Project administration:** Mark Y. Stoeckle, Greg Hinks, Stacy M. VanMorter.

**Resources:** Greg Hinks, Stacy M. VanMorter.

**Visualization:** Mark Y. Stoeckle.

**Writing – original draft:** Mark Y. Stoeckle.

**Writing – review & editing:** Mark Y. Stoeckle, Jesse H. Ausubel, Greg Hinks, Stacy M. VanMorter.

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
