## [Decision Letter · Decision Letter 0]

29 Sep 2024

PONE-D-24-36154A potential tool for marine biogeography: eDNA-dominant fish species differ among coastal habitats and by season concordant with gear-based assessmentsPLOS ONE

Dear Dr. Stoeckle,

Thank you for submitting your manuscript to PLOS ONE. After careful consideration, we feel that it has merit but does not fully meet PLOS ONE’s publication criteria as it currently stands. Therefore, we invite you to submit a revised version of the manuscript that addresses the points raised during the review process.

I have two reviews, both recommend acceptance of the manuscript. The first review has several constructive suggestions . I urge the authors consider the comments offered and revise suitably 

We look forward to receiving your revised manuscript.

Kind regards,

Arga Chandrashekar Anil, Ph. D., D. Agr.,

Academic Editor

PLOS ONE

Journal requirements: 1. When submitting your revision, we need you to address these additional requirements.Please ensure that your manuscript meets PLOS ONE's style requirements, including those for file naming. The PLOS ONE style templates can be found at https://journals.plos.org/plosone/s/file?id=wjVg/PLOSOne_formatting_sample_main_body.pdf and https://journals.plos.org/plosone/s/file?id=ba62/PLOSOne_formatting_sample_title_authors_affiliations.pdf 2. Please ensure that you refer to Figure 7 in your text as, if accepted, production will need this reference to link the reader to the figure. 3. Please include a caption for figure 1, Fig 2, Fig 3, Fig 4, Fig 6, Fig 8. 4. We note that the grant information you provided in the ‘Funding Information’ and ‘Financial Disclosure’ sections do not match.  When you resubmit, please ensure that you provide the correct grant numbers for the awards you received for your study in the ‘Funding Information’ section. 5. We are unable to open your Supporting Information file [S1_file.fas]. Please kindly revise as necessary and re-upload.

Reviewers' comments:

Reviewer's Responses to Questions

**Comments to the Author**

1. Is the manuscript technically sound, and do the data support the conclusions?

Reviewer #1: Yes

Reviewer #2: Yes

2. Has the statistical analysis been performed appropriately and rigorously? 

Reviewer #1: Yes

Reviewer #2: Yes

3. Have the authors made all data underlying the findings in their manuscript fully available?

Reviewer #1: Yes

Reviewer #2: Yes

4. Is the manuscript presented in an intelligible fashion and written in standard English?

Reviewer #1: Yes

Reviewer #2: Yes

5. Review Comments to the Author

Reviewer #1: I commend the work of Stoeckle et al for continuing to explore how eDNA tools can be applied to quantitative analysis of fish stocks. This is really important work that few people in the eDNA community seem to want to tackle head on.

In this most recent paper the team explores how ‘hollow curves’ can be used to interweave different types of data. The theory behind this is old but I think it is a really important way to analyse these types of data.

I recommend the paper for publication as we need eDNA researchers to push boundaries and provide some new thinking around how eDNA quant data can be used in different ways.

I have a few suggestions for the paper below, none of which are ‘deal breakers’ but might help the authors in their revisions.

#1 Introduction. In my mind there is not enough background on quantitative eDNA in the introduction and how there are different modalities (presence/absence; rank abundance and absolute abundance). This could be a segway into the tension when designing a new ruler for measuring fish – namely that the historical

methods are not easy to dovetail. The hollow curve methods and eDDS approach is a way to approach this.

#2 Discussion. There is an opportunity in the discussion and abstract to make a few valid points on fisheries management. While many in eDNA are sweating the small stuff (the tail of the hollow curve) there is a real possibility that we are missing the big picture. This eDDS approach is one tool to track the bigger picture and thus act as a triage for big changes geographically or seasonally.

#3 Biobanks. The paper should (I think) touch upon the fact that the eDNA samples provide an archive that can be reinterrogated at later points. As eDNA builds over time more intricate models could come into play that might require more (or different) assays.

#4 I am not advocating more experiments but one weakness of this approach is the reliance on a single fish assay. I guess I am asking at what point is it appropriate to interweave more than one assay into analyses such as this. Would a second assay provide reassurance?

#5 Why draw the line at 10 taxa for eDDS. It would be good to explore what happens if this number was extended? I think it is valid to ask this question as this cut-off seems quite arbitrary (as acknowledged line 286). While the number 10 might be useful for the NE fisheries this cut off might not perform as well in other habitats (e.g. coral reefs). Is there a heuristic was to pick a cut-off?

#6. I know there are already a lot of figures in the manuscript but the correlations in figs S3,4,5 are worth trying to bring into the paper – they are an important piece of the puzzle. To make space you could combine the seasonal figs 7+8 into a single figure.[as an aside the title of the paper in the supp info is different from that on the main paper]

#7: A leave one out sensitivity analysis would be really useful here – your ability to assign to ‘regional profiles’ or to ‘seasons’ is a really innovative part of this study but is (in my view) is underbaked. If I could suggest one extra analysis it would be to do this.

#8 In the conclusion if you are going to invoke ecosystem-based metrics from eDNA you should mention other assays that can profile other parts of the foodweb.

Thanks again for the chance to review this paper which I think is really important tool to add to the eDNA toolkit.

Reviewer #2: The authors describe a study focusing on comparing eDNA and net catch results from common species, which make up the vast majority of reads and biomass in the respective survey types. The paper is well written and the methods and analyses are appropriate. The only tiny comment I have is that the authors refer to the paper as a report. Maybe change that to manuscript or study or something. Otherwise I think this paper will make a nice contribution to the field and I recommend that it be published.

6. PLOS authors have the option to publish the peer review history of their article (what does this mean?). If published, this will include your full peer review and any attached files.

Reviewer #1: No

Reviewer #2: No

---

## [Author Response · Author response to Decision Letter 0]

15 Oct 2024

PLOS ONE

Academic Editor

Dear Dr. Anil,

Thank you for the careful reading and valuable comments by yourself and the reviewers on our manuscript. 

Regarding Journal requirements identified in your letter:

1. PLOS ONE style requirements. We believe the revised manuscript follows PLOS ONE style templates. 

2. Former Figure 7 (now Fig 8) is referred to in the text line 280 [“…(Figs 7 and 8)”].

3. Captions are included for all figures (note new Fig 5 added to address Reviewer 1 comment): 

 Fig 1, Line 114 (title, no legend)

 Fig 2, Line 202 (title, legend)

 Fig 3, Line 211 (title, legend)

 Fig 4, Line 228 (title, legend)

 Fig 5, Line 233 (title, legend)

 Fig 6, Line 249 (title, legend) 

 Fig 7, Line 269 (title, legend)

 Fig 8, Line 291 (title, legend) 

 Fig 9, Line 294 (title, no legend) 

4. Grant information is corrected.

5. Supporting Information file S1 is re-formatted as a “.txt” file, can be opened with TextEdit or similar programs. 

6. Reference list is complete (6 new references added to address Reviewer 1 comments). No retracted papers are cited. 

Response to Reviewer comments are detailed in separate file. 

We hope our revised manuscript is suitable for publication in PLOS ONE. 

Sincerely, 

Mark Stoeckle

Response to Reviewer Comments

We thank the reviewers for careful reading and valuable comments on our manuscript. 

Reviewer 1. 

Comment #1: Introduction. In my mind there is not enough background on quantitative eDNA in the introduction and how there are different modalities (presence/absence; rank abundance and absolute abundance). This could be a segway into the tension when designing a new ruler for measuring fish – namely that the historical methods are not easy to dovetail. The hollow curve methods and eDDS approach is a way to approach this.

See after Comment #2 below.

Comment #2 Discussion. There is an opportunity in the discussion and abstract to make a few valid points on fisheries management. While many in eDNA are sweating the small stuff (the tail of the hollow curve) there is a real possibility that we are missing the big picture. This eDDS approach is one tool to track the bigger picture and thus act as a triage for big changes geographically or seasonally.

We agree and appreciate the encouragement to raise these larger issues, including potential integration with established surveys. Manuscript is revised as below, with new citations [16,17] addressing use of historical records and issues of uncertainty in fisheries surveys. 

Abstract (line 36): “Grouping abundant species by category facilitated comparisons among habitats and integration with established surveys.” 

Introduction, line 57: “Marine eDNA investigations have often focused on presence/absence with an emphasis on detecting rarer species. Accumulating reports indicate fish species relative abundance is similar with eDNA methods including metabarcoding as compared to established census technologies [8-11]. Addition of copy number standards enables measuring eDNA absolute abundance, a further step towards censusing fish populations with eDNA data [12-15]. This progress raises the issue of how best to meld eDNA surveys with those obtained with other methods, one case of a general challenge for fisheries science [16,17].”

Discussion, line 342: “We suggest that the eDDS protocol described here, which takes advantage of the lopsided nature of SADs, is suitably accurate in present form for assessing dominant species. The protocol potentially offers a way to screen eDNA and traditional datasets for large-scale differences in common species by geography, season, or over time.”

Comment #3 Biobanks. The paper should (I think) touch upon the fact that the eDNA samples provide an archive that can be reinterrogated at later points. As eDNA builds over time more intricate models could come into play that might require more (or different) assays.

 We agree and have incorporated these observations. 

Discussion, line 346: “Looking ahead, biorepositories of archived eDNA samples, advances in DNA methods, and more complex modelling than outlined here may yield greater utility in ocean biomonitoring.” 

Comment #4 I am not advocating more experiments but one weakness of this approach is the reliance on a single fish assay. I guess I am asking at what point is it appropriate to interweave more than one assay into analyses such as this. Would a second assay provide reassurance?

This is an important issue. Discussion is revised to address this comment, including new references to fish metabarcoding primer comparisons and to recent work quantifying eDNA metabarcoding to predict trawl catches. 

Discussion, line 336: “More generally, all fish metabarcoding primers have some degree of amplification bias; analyzing samples with multiple primer sets is often advised to add confidence in species detection and abundance assessments [47]. Guri and colleagues recently demonstrated that species-specific adjustments that take into account PCR bias and fish catchability enable close prediction of trawl catches from eDNA metabarcoding data [14]. At present, wide usage of this approach may be limited by need to determine PCR and catchability adjustment parameters for each species of interest in a particular environment. We suggest that the eDDS protocol described here, which takes advantage of the lopsided nature of SADs, is suitably accurate in present form for assessing dominant species. The protocol potentially offers a way to screen eDNA and traditional datasets for large-scale differences in common species by geography, season, or over time.”

Comment #5 Why draw the line at 10 taxa for eDDS. It would be good to explore what happens if this number was extended? I think it is valid to ask this question as this cut-off seems quite arbitrary (as acknowledged line 286). While the number 10 might be useful for the NE fisheries this cut off might not perform as well in other habitats (e.g. coral reefs). Is there a heuristic way to pick a cut-off?

We think that evaluating other arbitrary cutoffs is beyond the scope of the paper, although we have looked at this informally. It tends to reduce categorical differences between habitats or seasons, as species abundant in one set are often present at a lower level in the other set. We agree that heuristic alternatives to top 10 is an important issue, this is added to discussion.

Discussion, Line 320: “Designating “top ten” as eDDS is an arbitrary cut-off that incorporates rare taxa in some datasets and not others. Potential alternative heuristics defining common species include the rank-ordered set comprising greater than 90% of total eDNA copies, or all individual species comprising greater than 1% eDNA copies.”

Comment #6. I know there are already a lot of figures in the manuscript but the correlations in figs S3,4,5 are worth trying to bring into the paper – they are an important piece of the puzzle. To make space you could combine the seasonal figs 7+8 into a single figure.[as an aside the title of the paper in the supp info is different from that on the main paper]

Figs S3,S4 are moved to main manuscript as new Fig 5. Regarding Fig S5, we respectfully prefer to leave this in SI, as the relationship between reads and copies was examined in greater detail with more context in a prior paper (Stoeckle et al. Environ DNA. 2022;6:e376.)

Comment #7: A leave one out sensitivity analysis would be really useful here – your ability to assign to ‘regional profiles’ or to ‘seasons’ is a really innovative part of this study but is (in my view) is underbaked. If I could suggest one extra analysis it would be to do this.

We appreciate this important recommendation. Manuscript is revised as follows: 

Materials and methods, line 173: “Leave-one-out meta-analysis was performed by omitting one sample or seine day at a time from each habitat or seasonal eDNA dataset then re-calculating the eDDS profile based on truncated set.”

Results, line 196: “As described in Materials and Methods, leave-out-one meta-analysis assessed contribution of individual samples/seine days to eDDS habitat profiles. The top ten taxa were identical to those in full dataset in most cases [16/18 (89%), 13/13 (100%), and 11/18 (61%), in sandy, rocky, and estuary datasets, respectively]; exceptions differed at 10th most common taxa.”

Results, line 264: “Leave-out-one meta-analysis assessed contribution of individual samples to eDDS seasonal profiles based on reads. The top ten taxa in truncated datasets were identical to those in full dataset in most cases [13/20 (65%) and 14/18 (78%) in summer and winter datasets, respectively]; exceptions differed at 10th most common taxa.”

Comment #8 In the conclusion if you are going to invoke ecosystem-based metrics from eDNA you should mention other assays that can profile other parts of the foodweb.

We agree, references are added to work analyzing plankton and potential food web interactions using eDNA and morphological tools. 

References (line 573):

“66. Bucklin A, Batta-Lona P, Questel J, McMonagle H, Wojcicki M, Llopiz J, et al. Metabarcoding and morphological analysis of diets of mesopelagic fishes in the NW Atlantic Slope water. Front Mar Sci. 2024;11:1411996.

67. Djurhuus A, Closek CJ, Kelly RP, Pitz KJ, Michisaki RP, Starks HA, et al. Environmental DNA reveals seasonal shifts and potential interactions in a marine community. Nat Commun. 2020;11:254.” 

Reviewer 2. 

The only tiny comment I have is that the authors refer to the paper as a report. Maybe change that to manuscript or study or something. 

In all cases referring to current submission, the word “report” is changed to “paper”, “study”, or “investigation”.

---

## [Editor Report · Decision Letter 1]

21 Oct 2024

A potential tool for marine biogeography: eDNA-dominant fish species differ among coastal habitats and by season concordant with gear-based assessments

PONE-D-24-36154R1

Dear Dr. Stoeckle,

We’re pleased to inform you that your manuscript has been judged scientifically suitable for publication and will be formally accepted for publication once it meets all outstanding technical requirements.

Kind regards,

Arga Chandrashekar Anil, Ph. D., D. Agr.,

Academic Editor

PLOS ONE
---

## [Editor Report · Acceptance letter]

30 Oct 2024

PONE-D-24-36154R1 

PLOS ONE

Dear Dr. Stoeckle, 

I'm pleased to inform you that your manuscript has been deemed suitable for publication in PLOS ONE. Congratulations! Your manuscript is now being handed over to our production team.

Kind regards, 

on behalf of

Professor Arga Chandrashekar Anil 

Academic Editor

PLOS ONE